# Multi-disciplinary Evaluation of Sexual Assault Referral Centres (SARCs) for better Health (MESARCH): protocol for a 1-year cohort study examining health, well-being and cost outcomes in adult survivors of sexual assault attending SARCs in England

Lorna O'Doherty,[1] Grace Carter,[1] Eleanor Lutman-White,[1] Rachel Caswell ,[2] Louise J Jackson ,[3] Gene Feder ,[4] Jon Heron,[5] Richard Morris,[6] Katherine Brown [7]

For numbered affiliations see end of article.

**Correspondence to**
Dr Lorna O'Doherty;
lorna.odoherty@coventry.ac.uk

## ABSTRACT

**Introduction** Sexual violence is commonplace and has serious adverse consequences for physical and mental health. Sexual Assault Referral Centres (SARCs) are viewed as a best practice response. Little is known about their effectiveness and cost-effectiveness. Long-term data on the health and well-being of those who have experienced rape and sexual assault are also lacking.

**Methods and analysis** This is a mixed-methods protocol for a 1-year cohort study aiming to examine the health and well-being in survivors of sexual violence after attending a SARC in England. Quantitative measures are being taken at baseline, 6 and 12 months. Post-traumatic stress (PTS) is the primary outcome (target N=270 at 12-month follow-up). Secondary measures include anxiety, depression, substance use and sexual health and well-being. Using mixed-effects regression, our main analysis will examine whether variation in SARC service delivery and subsequent mental healthcare is associated with improvement in trauma symptoms after 12 months. An economic analysis will compare costs and outcomes associated with different organisational aspects of SARC service delivery and levels of satisfaction with care. A nested qualitative study will employ narrative analysis of transcribed interviews with 30 cohort participants and 20 survivors who have not experienced SARC services.

**Ethics and dissemination** The research is supported by an independent study steering committee, data monitoring and ethics committee and patient and public involvement (PPI) group. A central guiding principle of the research is that being involved should feel diametrically opposed to being a victim of sexual violence, and be experienced as empowering and supportive. Our PPI representatives are instrumental in this, and our wider stakeholders encourage us to consider the health and well-being of all involved. We will disseminate widely through peer-reviewed articles and non-academic channels to maximise the impact of findings on commissioning of services and support for survivors.

## STRENGTHS AND LIMITATIONS OF THIS STUDY

⇒ The study proposes the most substantial investigation to date in the UK of health, well-being and service utilisation in survivors of sexual assault and rape.

⇒ The study applies a prospective, longitudinal design with a national sample of adults in England taking up Sexual Assault Referral Centres (SARC) services following experience of recent or non-recent sexual assault or rape, including all genders, to look at outcomes over 1 year.

⇒ The study considers a wider definition of sexual health than just sexually transmitted infection acquisition.

⇒ The study uses innovative approaches to recruitment that address barriers to research participation in the target population, including challenges created by COVID-19 and is supported by a wide range of stakeholders nationally including a group of people with lived experience of sexual violence and abuse.

⇒ Relatively few survivors of sexual assault and rape have used SARCs, so generalisability to all people who have been exposed to sexual violence and abuse may be limited.

**Trial registration number** ISRCTN30846825.

## INTRODUCTION

Sexual violence is commonplace, with evidence suggesting that one in five women and one in twenty-five men in England and Wales have experienced sexual assault since the age of 16.[1] Sexual violence refers to any sexual act, or attempt at a sexual act, or an act directed at a person's sexuality,

involving coercion.[2] Sexual violence includes but is not restricted to: rape, sexual assault, child sexual abuse, sexual harassment, rape within marriage and relationships, forced marriage, so-called honour-based violence, female genital mutilation, trafficking, sexual exploitation and ritual abuse.[2] Sexual violence often goes unreported to the police, though recent shifts in the public understanding of sexual violence and developments that include the *#metoo* movement on social media have likely contributed to year-on-year increased rates of reporting.[3] Prior to the COVID-19 pandemic, fewer than 1 in 6 victims of rape reported to the police. In the year ending June 2021, the number of sexual offences recorded by the police (164 763 offences) showed an 8% increase compared with the previous year.[4]

The serious and devastating effects of sexual violence are a well-documented public health burden.[5] They include a range of immediate and long-term physical and mental health consequences. For women who have experienced sexual violence, physical health consequences include unwanted pregnancy, sexually transmitted infections (STIs),[5] urinary tract infections, painful sex, chronic pelvic pain and vaginal bleeding.[6] For male victims, physical health consequences include genital and rectal injuries and erectile dysfunction.[7] The mental health sequelae of sexual violence have been found to be equally substantial across different population groups.[5 7–9] Post-traumatic stress (PTS) is common among those who have experienced sexual violence,[10 11] with incidence and severity being similar in men and women.[12] Other mental health consequences include alcohol use disorders, eating disorders, anxiety, depression, self-harm and suicidality.[5] In addition, experiencing mental health problems such as lifetime PTS is associated with increased risk of other long-term health conditions including hypertension, cardiovascular disease and gastrointestinal problems.[10] Thus, sexual violence produces significant health burden, with the wider economic costs to society reaching over £12 billion per year.[13]

Given the substantial personal, societal and economic costs of sexual violence, an effective and consistent response for survivors is imperative. In the 2003 guidelines on the provision of care for victims of sexual violence, the WHO recommended that an initial response to survivors should include medico-legal and health services provided at the same time, in the same location and preferably by the same health practitioner. Policy-makers, commissioners and health workers have been encouraged to develop this model of service provision.[14] In the USA for example, many states offer Sexual Assault Nurse Examiner programmes or Sexual Assault Response Teams to provide these recommended services. In England and other parts of the UK, there has been a steady growth in Sexual Assault Referral Centres (SARCs)

as a best practice response for survivors after incidents of sexual assault or rape.[15] SARCs, as a form of medico-legal and health service, are intended to coordinate all of the care and support needs for survivors. This may include crisis emotional support, forensic medical examination for the purposes of collecting evidence needed to prosecute alleged perpetrators, provision of emergency contraception and HIV Post-Exposure Prophylaxis, referral to sexual health and other healthcare services, referral for mental health-care needs (eg, counselling and therapy) and to an independent sexual violence advisor (ISVA) for ongoing support, particularly if dealing with the criminal justice system. There are approximately 50 SARCs in England including a number of specialist paediatric sites,[16] operating with considerable variation in their service models.[17] Currently, little is known about the effectiveness and cost-effectiveness that the different service models may represent in addressing the physical and mental health outcomes for survivors of sexual violence outlined above. In addition, although the substantial negative impacts of sexual violence are well-documented[5] much less is known about the longer-term health and well-being outcomes for survivors. One exception is a recent study in South Africa which followed up female rape survivors over 3 years and, controlling for confounding variables, has shown they had a 60% increased risk of contracting HIV in that time compared with control group women who had not been raped.[18–20] Data related to models of care survivors receive and health outcomes remain limited, however. Similarly, there is a paucity of data on (mental) healthcare access after SARC[21] and understanding of how different service user characteristics might moderate the benefits of attending SARC (and other specialist services for sexual assault and rape), in particular characteristics associated with social inequities and marginalisation (eg, considering service users of migrant and ethnic minority backgrounds, and sexual and gender minority communities). Furthermore, there is a lack of data on the experiences of male survivors[22]; those with a disability[21] even though they are at greater risk of sexual violence[23]; interactions between sexual violence exposure and chronic mental health problems, and comorbidity of health outcomes; non-partner sexual violence[24] and sexual health outcomes other than the acquisition of sexually transmitted infections (STIs).[25]

This mixed-methods cohort study is the first to consider the health outcomes and cost trajectories of those who attend SARCs in England. It aims to evaluate the relationship between SARC factors, subsequent care (follow on specialist sexual assault services and mental healthcare), and trajectories of PTS; the relationship between participant characteristics and trajectories of PTS; and finally, the role that participant characteristics may have in moderating the benefits of different SARC factors and subsequent care. It

represents the first such study to consider the broader sexual health and well-being of survivors attending SARCs alongside other assessments of their health and mental well-being over time. We also report here how findings from our pilot study at one English SARC and our early data collection experiences have informed a revised research protocol.

## METHODS AND ANALYSIS
### Design and setting
This is a mixed-methods cohort study of mental, physical, sexual health and cost outcomes over 1 year in adult survivors of sexual assault and rape who have received care through SARCs in England. Quantitative measures are taken at baseline (no sooner than 4 weeks) and at 6 and 12 months post-baseline. A nested qualitative study is included (see the Qualitative study section). The study is part of the Multi-disciplinary Evaluation of SARCs for better Health (MESARCH) programme (1 September 2018 to 31 January 2023). The wider programme includes two Cochrane Reviews,[26 27] and 'mapping' English SARCs, their structure and services, in addition to the study described here.

### Sampling, recruitment and procedures
We have recruited a stratified random sample of 15 of all the current practicing adult SARCs in England (approximtely 50 identified at the time of writing) with strata defined according to service delivery model, size and level of integration of services. To do this, we have drawn on national indicators of performance data collected by National Health Service (NHS) England and NHS Improvement,[15] the commissioning body for SARCs, and data collected from SARCs as part of the wider MESARCH study.

Recruitment is taking place in three phases. First, service users aged 18 years and above presenting in person (or remotely since COVID-19) at any of the 15 participating SARCs are screened by SARC staff. People are excluded if, in exercising own judgement, the responsible member of SARC staff conducting screening anticipates a service user may encounter difficulties in providing informed consent due to severe mental health issues, cognitive impairment or learning disability. We also exclude those visiting from prison settings, as conducting follow-up assessments in prisons is outside the scope of the project. SARC staff complete a screen form for all those over 18 presenting to the service, recording basic demographic data, and offence and referral characteristics and indicating whether the person is eligible or not, with any reasons for exclusion noted. These screen forms are given an identification number, contain no identifying information, and are passed to the project team to provide an indication of the characteristics of the pool of service users pre-enrolment.

Because many SARCs refer on quickly to other voluntary sector agencies (ie, non-governmental organisations

(NGOs)), for each participating SARC, we identified all linked support agencies and worked to engage them. The second phase of recruitment mainly takes place in that follow-on setting. When the person is referred to follow-on services, the identification number assigned at SARC travels with them. The study is introduced in that setting by trained staff from the agency, and if the person/service user expresses an interest in hearing more about the study, the staff member seeks consent to pass on contact details to the researchers. The identification number is also passed to the research team at this point, thus allowing the researchers to link the person/contact details with the SARC data obtained in the first phase.

In the final phase, the research team takes responsibility for contacting service users and inviting them to participate. Box 1 provides further detail on the recruitment procedure. SARC service users may also self-refer to the study or be referred to the study directly by SARCs. Anyone aged 18 years or over who has visited a SARC in England during the recruitment phase (2 years) since 1 August 2019 was eligible for enrolment via direct contact with the research team. Please note a separate study focused on those aged 13–17 years is also being conducted as part of the wider MESARCH programme.

### Number of participants required
At each of the 15 participating SARCs and via some self-referral, eligibility and basic sociodemographic data are collected on a total of 2500 individuals enabling us to compare our cohort sample with the wider population of service users at SARC. The primary outcome of interest for evaluating service user journeys after attending a SARC is PTS, measured using the PCL-5.[28] Our interpretation of models based on the PCL-5 will focus on what constitutes a minimal clinically important difference (MCID) for PTS. Stefanovics and colleagues[29] concluded that differences of approximately 0.5 of an SD could be deemed clinically important both cross-sectionally and for within-person change. Since our study involves neither randomisation nor placebo-control, our analyses will focus on change in PTS symptoms between baseline and follow-up. We will take recommendations of Stefanovics et al[29] as quantifying the change-score group-difference we would expect which separates a treatment that is effective from one that is ineffective. Based on guidelines,[28] we anticipate this MCID of 0.5 SD to approximate to a 10-point difference on the PCL-5 change-score; however, Stefanovics et al[29] propose that an SD-based approach would be more robust and we also recognise that the magnitude of change in on the raw-scale metric will be influenced by the length of time between baseline and follow-up.

To allow for four levels in a predictor variable (eg, four types of SARC service provider); up to 10 covariates; two time points; an expected small-to-moderate Cohen's f, which corresponds to the MCID for our primary outcome variable[29]; and an α error probability of 0.05, 270 participants would be required at study end to achieve a power (1−β error probability) of 0.90. Based

**Box 1   Procedure for involving adult service users in the Multi-disciplinary Evaluation of Sexual Assault Referral Centres (SARCs) for better Health (MESARCH) cohort study**

1. Service user receives care from the SARC.
2. SARC staff member screens service user for eligibility. All eligibility information is de-identified (assigned an identification number only) and conveyed to the project team using secure sharing platform.
3. A referral is made by SARC to an independent sexual violence advisor (ISVA; usually based within a voluntary sector agency) for follow-on care in line with usual practice, and the research identification number is placed on the referral form. If the service user is not willing to be referred to other agencies, the SARC could provide more information about the study at this point.
4. Within the voluntary sector agency, eligible service users are approached mainly by ISVAs, who provide brief explanation about the project and invite service users to consider being involved in the study. For those who express an interest in hearing more, consent is requested to pass contact details to the project team and the person's research identification number is also included ('Level 1' consent completed).
5. Additional approaches for enabling recruitment include (a) having members of the project team ready to speak by phone or video link to a service user if the SARC worker/ISVA believes this is an appropriate approach for a particular service user; (b) showing a short video message of invitation co-produced with our Lived Experiences Group which could be used by SARC staff/ISVA to explain about the project (available on our project website and accessible using a QR code on study brochure); (c) opening study up to self-referrals by survivors who have used SARCs in England during the study recruitment phase.
6. Once the contact information has been passed on to the researchers, there is no more involvement of the SARC/ISVA except in circumstances such as the participant requires support and has agreed for us to notify the SARC/ISVA.
7. A trained project team member makes contact within 1 week of receiving the research referral consent form from the SARC or ISVA, and contact is made in line with service user preferences for example by email, text or phone call.
8. Once contact is made by the project team, the project team will follow recruitment and safety protocols, explain study purpose and gain full consent ('Level 2' consent).
9. When consent to take part in the study is established (using email or text or signed consent form), baseline data are collected using a range of options. These include offering a structured telephone interview, a weblink to complete the data collection online, an interview in person or via video link. Those submitting baseline data will be considered 'enrolled' in MESARCH.
10. Follow-up is undertaken according to the participants' preferences at 6 and 12 months consistent with our previous work.[50]

on previous studies,[19 20] our pilot work, and recruitment and retention to date, we estimate attrition at 20% by 12-month follow-up, requiring a baseline target of 338 individuals (see estimated flow of participants in figure 1). Participants who do not complete a 6-month follow-up will still be eligible to complete 12-month follow-up measures.

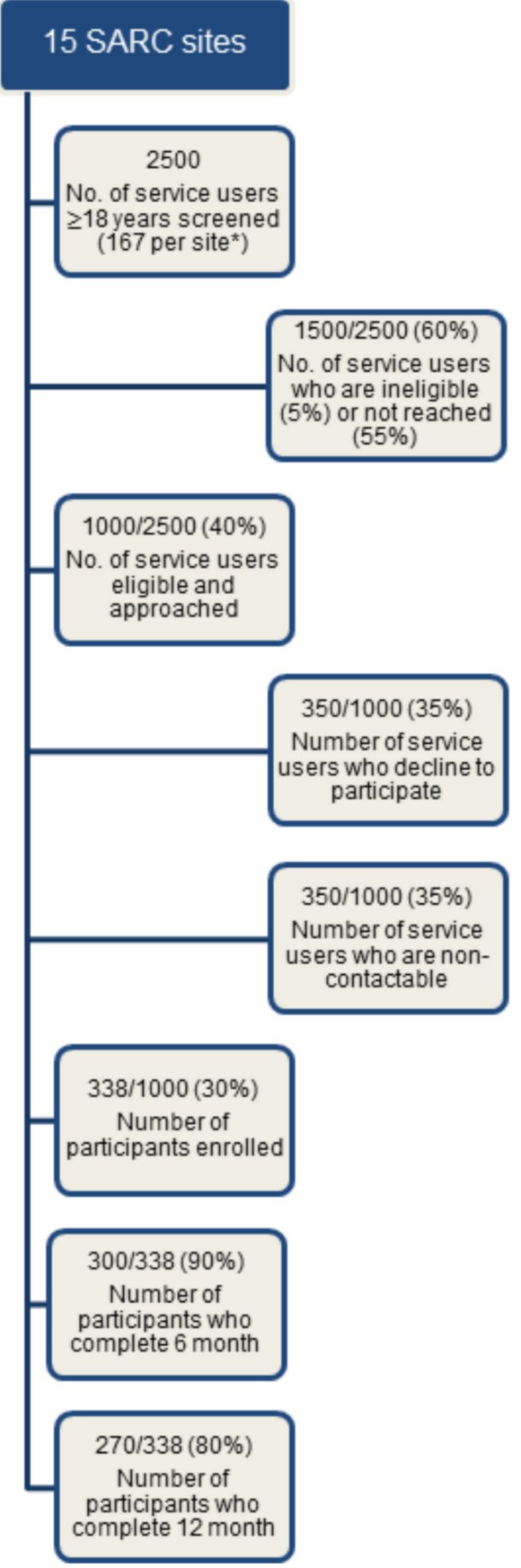

**Figure 1**   Flowchart of service users aged 18 years and above in the Multi-disciplinary Evaluation of Sexual Assault Referral Centres (SARCs) for better Health cohort study.

## Data collection

The form used by SARC staff to screen service users for the research provides data on gender; age; ethnicity; offence characteristics; referral into SARC and indicators of vulnerability at the point of service access (eg, history of mental health problems, disability, substance use). At baseline, we collect additional detailed sociodemographic information on marital/relationship status; nationality; immigration status; number of children; sexual identity; disability; religion; education; employment status and income; and accommodation. Baseline data is gathered using structured interviews, by telephone, video link or in-person by trained research staff who enter responses directly into secure bespoke study software linked to Qualtrics survey software. We enquire about adverse childhood experiences using the WHO ACE International Questionnaire and screen for lifetime and past-year domestic abuse using the ACTS[30] to enable us to describe the sample. All other baseline measures focus on standardised timeframes according to the measure, or the period since participants attended the SARC. Follow-up is taking place on two occasions over the course of the project, at 6 months and 12 months. Our proposed methods for retaining participants are informed by a number of large studies of violence and health[31] including our own previous work in the health field.[32] These include gathering a range of contact details from participants at baseline - safe telephone numbers; postal/email addresses; use of reminders; and providing a small incentive (ie, £10 voucher following completion of baseline and 6 months and £15 voucher at 12 months).

## Outcomes

Table 1 shows the measures at each time point. The primary outcome is PTS, widely endorsed in the literature[5 10] and through our scoping work with individuals with lived experience and service providers, as a primary health issue for survivors and an absence of diagnosis, or reduction in symptoms, may mark improvement or recovery in a person who has experienced sexual violence. Our analyses will focus on change or improvement in PTS symptoms between baseline and follow-up. We have developed a set of items to assess sexual health, including STI diagnosis; pregnancy and pregnancy outcomes, and reproductive coercion.[33 34] Though we screen for sexual violence re-exposure at follow-up, discussion of the experience of sexual violence that led the person to seek help at the SARC is not encouraged in order to protect victims' testimonies and to create a safe space for the person to participate in the research. Our approach to measuring health and other costs using the ICECAP-A[35] and EQ5D-5L[36] is outlined in the Economic evaluation section. We have also developed a comprehensive evaluation of SARC service users' onwards service utilisation and satisfaction with care that considers the many sectors that are implicated in meeting the needs of survivors in the aftermath of an assault or abuse.

## Patient and public involvement

The funding application for this research was made in response to a commissioned call from the National Institute for Health Research (NIHR) which set out a series of research questions concerning the evaluation of SARCs in England. We met with survivors of sexual violence as well

**Table 1** Outcome measures and the timepoint at which they are collected

| Outcome | Measure | Timepoint |
|---|---|---|
| Post-traumatic stress | PCL-5[28] | Baseline, 6 months, 12 months |
| Depression | Centre for Epidemiologic Studies-Depression Scale (CESD-R-10)[51] | Baseline, 6 months, 12 months |
| Quality of life | WHOQoL-Bref[52] | Baseline, 6 months, 12 months |
| Eating issues | SCOFF[53] BEDS-7[54] | Baseline, 6 months, 12 months |
| Health-related quality of life | EQ5D-5L[36] | Baseline, 6 months, 12 months |
| Capability | ICEpop CAPability measure for Adults (ICECAP-A)[35] | Baseline, 6 months, 12 months |
| Sexual health | Bespoke measure | Baseline, 6 months, 12 months |
| Alcohol use | AUDIT-C[55] | Baseline, 6 months, 12 months |
| Drug use | DUDIT[56] | Baseline, 6 months, 12 months |
| Resource use | Bespoke measure | Baseline, 6 months, 12 months |
| Adverse childhood experiences | WHO ACE International Questionnaire[57] | Baseline |
| Intimate partner violence | ACTS 4 item screen[30] Composite Abuse Scale (CAS- Short Form)[58] | Baseline 6 months, 12 months |
| Re-exposure to sexual violence | Bespoke measure | 6 months, 12 months |
| Suicidality and self-harm | Items from the Adult Psychiatric Morbidity Survey[59] | Baseline, 6 months, 12 months |

PTS, post-traumatic stress.

as other stakeholders (eg, rape crisis services, SARC staff) and consulted them throughout the funding application process to ensure that the proposed research met the diverse needs of survivors and providers, and was acceptable and feasible from their perspectives. When funding was awarded, we formalised Patient and Public Involvement (PPI) by recruiting a group of survivors to support the study on an ongoing basis. Members named the PPI group as the Lived Experiences Group (LEG) and it is made up of 10 survivors from different backgrounds and experiences. Two members of the LEG sit on our study steering committee (SSC) and the wider group has been involved in every aspect of our study decision-making, the design of our materials, our communications about the study and production of outputs, development of follow-on projects. The LEG will continue to support the process of understanding and interpreting our findings, shaping the recommendations we make, and disseminating effectively with all stakeholders, including our participants, about what we find.

## Data analysis

Our analysis will determine if variation in SARC service delivery and other healthcare specific to sexual violence is associated with different levels of improvement in trauma symptoms between baseline and the 12-month follow-up. Our favoured statistical approach will be to model 'trajectories' of PTS using mixed-effects regression (ie, multilevel modelling, using Stata's suite of mixed-effects tools[37]). This framework will enable us to model variation in the improvement in PTS and make best use of the three waves of data available. In the eventuality that change appears non-linear, we will first explore the utility of incorporating varying times of follow-up and modelling time-since-baseline rather than data-collection-wave as the time axis, although we will be mindful of the potential for introducing bias by incorporating such a source of variation. We will also consider standard single-level regression models (eg, Ordinary Least Squares (OLS) regression, Analysis of Covariance (ANCOVA)) and model observed change in PTS between baseline and 12 months, instead using the 6-month wave to assist with the treatment of any missing data using multiple imputation. In addition, and where appropriate, we will accommodate the nested data structure of participants within SARC-centres using cluster-robust SEs. Similar analyses will be carried out for the secondary outcomes.

(1) Following the investigation of the optimal 'unconditional' model to describe changes in PTS and the other outcomes of interest, we will introduce SARC-level characteristics (whether ISVAs are located at SARC or externally provided; whether the SARC is run by police, NHS, voluntary agency or private company; regional capacity for meeting survivors' needs; ease of access to SARC) and characteristics related to follow-on care for sexual violence and mental health (uptake of ISVA care, access to therapy in the NHS or elsewhere), as a series of fixed effects. (2) The next stage of analysis will seek to address the question 'How do trajectories compare for different subgroups of survivors attending SARCs?'. In the event that the mixed models described earlier can be empirically justified, we will estimate outcome trajectories for different groups incorporating recent versus non-recent victimisation including time lag between trauma and SARC visit, ACEs, gender, sexuality, culture, ethnicity, disability and chronic mental health problems and perceived benefit/harm from services. Where necessary, the single-level equivalent for this analysis stage will examine whether mean differences in PTS improvement vary across the same participant characteristics. While this stage mentions 'subgroups', and technically involves an interaction (with time since baseline), this can be considered both conceptually, and in terms of statistical power, to be a series of main-effects analyses. (3) Finally, we will investigate the role that participant characteristics may have in moderating the benefits of different SARC factors and follow-on care for sexual violence and mental health. This analysis will be both exploratory and pragmatic given the likely low power for this step and will be driven by the findings from stages (1) and (2) above. We will focus on those minority groups large enough to enable robust statistical comparison and formal interactions tests to be performed. In instances where some subgroups of interest are small (eg, male survivors), hindering formal statistical tests, we will take a more descriptive approach and be cautious about our conclusions.

# ECONOMIC EVALUATION
## Design

The economic analysis will compare the costs and outcomes associated with different organisational aspects of SARC service delivery. If some SARC models are more strongly associated with reducing PTS and improving quality of life and other mental, physical and sexual health outcomes than others, there are likely to be important cost implications for the healthcare sector, for the wider public sector, and for society as a whole.

## Data collection

Resource use data will be collected prospectively to estimate the costs associated with different models of SARC service delivery. The resource use to be monitored will include: (1) the cost of service use within SARCs (eg, consultations, treatment, etc); (2) NHS and other public sector resource use after initial attendance at SARCs (eg, general practitioner visits, sexual health visits); (3) costs associated with the treatment of PTS and other relevant conditions; (4) wider public sector resource use, for example, in relation to social care, housing and other social welfare systems; (5) costs experienced by service users and family members. Information on unit costs or prices will be sourced to attach to each resource use item (eg, Curtis and Burns[38]). Health-related quality of life data will be collected using the EQ5D-5L instrument which is widely used for those with PTSD and related conditions.[39]

## Economic analysis

In order to compare the costs and benefits of different SARC service delivery models, both a within study analysis and a model-based economic analysis will be undertaken. The main economic analysis will assess cost-effectiveness based on incremental cost per quality-adjusted life year gained at 6 and 12 months, with a secondary analysis of cost per case of PTS avoided at 12 months. If the results of the cohort study show that there are significant differences in the effectiveness of SARC delivery models, in terms of reducing PTS and improving other health outcomes, it will be necessary to assess the cost-effectiveness of the SARC delivery models in the longer term, to take into account the impact on an individual's quality of life and productivity. Therefore, if deemed necessary, based on the results of the cohort study, we will use a decision-analytic model to evaluate the longer-term impacts of the different types of service delivery (for up to 5 years, if data allow). The model development process will use, as a starting point, other models developed for PTS and related conditions (eg, Mihalopoulos et al[40]). We will use both deterministic and probabilistic sensitivity analyses to explore the effects of inherent uncertainty in the estimates on the results.[41] For the longer-term analyses, discounting will be undertaken to reflect recommendations by National Institute for Health and Care Excellence (NICE).[42]

## PILOT STUDY

A pilot study was conducted with baseline recruitment over 3 months between April and June 2019 at one SARC site. The purpose was to examine the feasibility of the methods for the main cohort study and identify required design modifications. We assessed the feasibility of our approach to inviting participants into the study, data gathering procedures and retention (eg, rates and use of incentives). At the SARC, we discussed the project with staff members, built awareness about the rationale, explained inclusion/exclusion criteria and use of the screening form to record eligibility, and agreed on the most appropriate staff members to approach potential participants at different stages. Once set-up was complete, all eligible service users over a 3-month period were invited into the pilot project by SARC workers. The pilot included telephone interviews at baseline and one follow-up only, at 3 months. Data collection included a set of questions about the experience of being invited into and participating in the study.

Over 3 months, 43 service users attended the pilot SARC (classed as small relative to other English SARCs). Staff tended to be highly conservative in applying the eligibility criteria, voicing concerns about mental ill-health and overburdening the service users (n=15). Of the 28 service users identified as eligible, 11 lost contact with the SARC prior to being invited into the study and three declined to have their contact details passed on. Of the 14 people who agreed to be contacted by the research team, four could not be contacted and ten were successfully enrolled.

From those ten participants, there were no missing data and retention at 3 months was 100%. We encouraged think-aloud responses to items and requested feedback about people's experiences of the pilot. These were all positive comments. For example, 'I really enjoyed being a part of the study. It felt great to be able to take an unfortunate event and turn it into something positive and it's nice knowing that what I contributed will go on to help others' (23-year-old woman participant in pilot study).

We are aware from another similar study,[19 20] our own pilot study and our early recruitment experiences of the challenges in recruitment. For the main study, we are providing clearer guidance to staff about the exceptional circumstances under which a person should be identified as ineligible. We are aware of the circumstances of people's lives that make research participation difficult, but we believe they should, wherever possible, be given full choice around being involved or not. Rates of exclusion at the SARC level are continuously monitored during recruitment and further training and communication interventions are implemented where deemed necessary. We have continued to focus our efforts on minimising the numbers of individuals who are potentially 'lost' between SARC and the follow-up care in NGOs and other settings, with ISVAs being our primary and safest avenue for reaching potential participants. In addition to working with ISVAs, we have cascaded information across other practitioners in the voluntary sector (eg, counsellors) and sexual health services to promote self-referral. We regularly refer participants and potential participants to our up to date website where people can read about the origins and aims of the work; understand about the people carrying out the work, including the role of our LEG members; and access testimonials from other survivor participants. We offer options to participants wherever possible (eg, choice of gender of interviewer and speaking with a peer researcher; offering interview times out of hours and doing interview in parts if desired; providing overview of each section before any questions are asked enabling people to skip sections/questions; providing resources and follow-up where concerns arise). We expect such measures will enhance our retention and minimise missing data.

## QUALITATIVE STUDY
### Design

There will be a nested qualitative, interview-based study to enable a greater depth of understanding of the factors associated with better outcomes for survivors of sexual violence participating in the adult cohort study. It will include a community-based comparison sample. We will include around 50 individuals aged 18 years and above.

The study will employ narrative methods[43] as these provide participants with the opportunity to give their accounts of a particular experience, free of the assumptions of the interviewer or research team and empowering them to structure the stories of what happened,

and their meanings as understood by them. Narrative methods have been applied extensively in explorations of experiences relevant to sexual assault and rape[44–46] and lend themselves well to gaining insight into the ways in which people get to grips with potentially devastating disruptions to their everyday lives. For example, Becker's[47] work takes an in-depth narrative approach to considering experiences of five people who have had different major disruptive life events including illness and infertility.[47]

## Recruitment of participants

Participants who have participated in the adult cohort study will be recruited in order to gain insight into factors that were experienced positively and negatively from a range of narratives. This will provide the project team with rich data about experiences outside of SARC models of care that were valued, as well as those within SARC models. In collaboration with our LEG and professionals who support survivors, we will devise a sampling framework and detailed recruitment strategy that is sensitive to the needs of, and acceptable to, the target population. We propose that this will involve recruiting adults from the SARC cohort study after their 12-month measures are collected. We will aim to recruit around 30 cohort members in total and apply maximum variation sampling in order to over-represent service users whose voices are not typically included in this type of research (eg, in this context, gender, sexual, age and ethnic minorities; participants affected by issues such as homelessness, sex work, disability and long-term mental health difficulties).

## Recruitment of non-SARC service users

We will also seek to recruit people via voluntary sector partners and stakeholders who support survivors of sexual violence. This will ensure wider experiences are included in the data (approximately 20 individuals) and will help the research team to better understand barriers to SARC access among those offered a referral by other services but who have not used SARCs.

## Procedure

Participants who express an interest in being interviewed will be given time and support to prepare for the narrative interview having had a full explanation of what this involves and ensuring they are comfortable to provide their story about experience of services following exposure to sexual violence. Participants will be offered the choice of a face-to-face (where COVID-19 restrictions allow), online video conference, or telephone interview at a time and location convenient to them, with or without the support of a person of their choosing. The interview guide will be developed in partnership with survivors of sexual violence within our lived experience group. It is likely to include an opportunity for the participants to talk generally about themselves and how they are, and to discuss any concerns they may have about the narrative interviews they are engaging with. The interviewer will discuss any concerns and ensure that the participant

is happy to talk about their recovery journey. The interviewer will likely suggest the participant starts at an appropriate moment in discussions by saying that he/she/they are, 'interested in finding out about your experiences of accessing help or support and the impact on your health and well-being, after experiencing sexual violence' and, 'you can begin to tell us your experiences of receiving support and your recovery journey so far, at the point where you first sought help in relation to your experience of sexual assault, or wherever it feels right to start'. We will be clear through our participant information sheet and consent procedures that we are not asking participants to talk about the incident(s) that led them to seek help and support. Interviews will be audio-recorded with participants' permission and transcribed verbatim.

## Analysis of qualitative data

Data will then be subject to narrative analysis[43] to draw out the meanings ascribed to participants' experiences and to identify both unique elements and commonality or themes across experiences. This process will be led by KB, with support from LOD. Narrative analysis is divided into two distinct phases. The first is a descriptive phase which, following thorough reading and familiarisation with the transcripts, involves devising summaries of each narrative to pull out the key features and identify subplots as well as overarching story arcs. Similarity between different narrative summaries, as well as key differences, will also be identified at this stage to form the basis of a coding frame. In the second stage, a range of theoretical perspectives will be considered in order to interpret and make sense of the narratives and the coding frame. To achieve this we will work collaboratively with co-investigators and LEG members, considering the range of options and the ways in which they may or may not aid interpretation of the data.

## ETHICS AND DISSEMINATION

The pilot study and main cohort study have received ethical approval from Coventry University (pilot: P75698; main study: P86669) and NHS Research Ethics (pilot: 18/WM/0376; main study: 19/EM/0198). NHS Health Research Authority approval was also provided before data collection began.

The research is supported by an independent study steering committee (SSC), a data monitoring and ethics committee (DMEC) and a patient and public involvement (PPI) group, known as the LEG (see the Patient and public involvement section). The SSC has been in place since study set up and is made up of a range of relevant academic, government, non-government and independent agencies and experts who can offer guidance and support on the challenges of conducting the research throughout. One example of SSC and LEG input includes working with them to refine our choice of measures, in particular, appropriate ways to assess sexual violence exposure in this high-risk population.[48] A further example is

having worked with some of our LEG members on the development of short videos to introduce the study to professionals and survivors to support recruitment. Both of these films can be accessed on the study website at http://mesarch.coventry.ac.uk/getting-involved/. The DMEC is meeting three times throughout the cohort study (baseline, post 6 months and post 12 months) to assess data, analyses, and whether any evidence of harm should lead to a recommendation that the study ceases.

The experience, safety and well-being of survivors, participants and staff is of paramount importance in this research. All staff involved in delivering this cohort study and the wider research activities in the funded project have undergone specialist training from two voluntary sector organisations who work to support survivors of sexual violence, participate in a weekly team meeting and receive regular training and external clinical supervision. We have developed a detailed safety protocol to define clearly what the actions of staff should be if there are any safety, safeguarding or well-being concerns raised during research procedures. These are with regards to potential participants, participants and any children or other dependents, and staff in line with NIHR Policy on Preventing Harm in Research. A central guiding principle of the research is that being involved as a participant should feel like the opposite of being a victim of sexual violence, derived from by The Survivors' Charter developed by Survivors' Voices.[49] In other words, it should involve feeling empowered, respected, and in control at all times. In fact, in our experience of data collection to date, during the pilot and the first participants at baseline in the main cohort study, people report exactly that. They are happy to and want to participate in the research because it is an opportunity for them to proactively contribute something positive to improve services in the future. It is their decision to do so and they are in control of the answers and information they provide.

One issue that it has been particularly pertinent to consider is the possibility that data provided by participants involved in an investigation or pursuing a prosecution may be subject at any time to subpoena by the courts or requested by law enforcement. Participants are made aware of this before providing consent, and the focus of the research is not what happened during any alleged incident of sexual violence or abuse. Our safety protocol involves a requirement that staff remind participants about the limits of confidentiality at each time point, including subpoena of interview notes. The team is regularly trained in how to manage disclosure of new information pertaining to a case or which signals a risk to the welfare of a child or vulnerable adult.

Dissemination of the findings will be supported by our SSC and the LEG through a range of channels in the future. In addition to standard publication in peer-reviewed academic journals and at academic conferences, we will present findings at policy and practice-based conferences nationally and internationally. We will publish findings in lay language on our study website and produce other forms of lay publications targeted at all agencies and organisations with a remit for addressing sexual violence and its consequences in society. We will work with NHS England and NHS Improvement, commissioners of SARCs in England, to ensure the findings are translated appropriately into policy and practice.

**Author affiliations**
¹Institute for Health and Wellbeing, Coventry University, Coventry, UK
²Sexual Health and HIV Medicine, University Hospitals Birmingham NHS Foundation Trust, Birmingham, UK
³Health Economics Unit, University of Birmingham, Birmingham, UK
⁴Community Based Medicine, University of Bristol, Bristol, UK
⁵Centre for Academic Mental Health, University of Bristol, Bristol, UK
⁶School of Social & Community Medicine, University of Bristol, Bristol, UK
⁷Department of Psychology, Sports Science and Geography, University of Hertfordshire, Hatfield, UK

**Acknowledgements** We would like to acknowledge the contributions of other MESARCH study co-investigators Professor Sarah Brown, Dr Emma Sleath, Millie Gant (Juniper Lodge Sexual Assault Referral Centre) and Dianne Whitfield (Rape Crisis England and Wales) as well as Priya Tek Kalsi, the project's engagement and PPI co-ordinator. Thanks to Dr Cain Clark for advice on sample size. Finally, we are grateful to the survivors who participated in the pilot study and to members of our Lived Experiences Group, Study Steering Committee and Data Monitoring and Ethics Committee for all their contributions and support.

**Contributors** LOD conceived the study design with support and input from GF, RM, RC, LJJ and KB. RM paid particular attention to designing the analysis plan and LJJ to designing the health economics plan. LOD led the successful application for funding with contributions from GF, RM, RC, LJJ and KB. GC supported by ELW oversees recruitment and data collection and LOD oversees the preparation of data in readiness for analysis. RM drafted the original cohort study analysis plan which was updated by JH. GC leads the children and young people's study. KB leads the nested qualitative study. KB drafted the manuscript based on the approved study protocol V3.3 and all co-authors read, contributed to re-drafting and approved the submitted manuscript. All authors have agreed to be accountable for all aspects of the work.

**Funding** This work is supported by the National Institute for Health Research (Health Services & Delivery Research programme) grant number 16/117/04.

**Disclaimer** The views expressed are those of the authors and not necessarily those of the NIHR or the Department of Health and Social Care.

**Competing interests** None declared.

**Patient and public involvement** Patients and/or the public are involved in the design, conduct, reporting, and dissemination plans of this research. Refer to the Methods section for further details.

**Patient consent for publication** Not applicable.

**Provenance and peer review** Not commissioned; externally peer reviewed.

**ORCID iDs**
Rachel Caswell http://orcid.org/0000-0002-9246-2581
Louise J Jackson http://orcid.org/0000-0001-8492-0020
Gene Feder http://orcid.org/0000-0002-7890-3926
Katherine Brown http://orcid.org/0000-0003-2472-5754

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
