## [Reviewer comments · BMJ Open]

ARTICLE DETAILS

TITLE (PROVISIONAL)	Multi-disciplinary Evaluation of Sexual Assault Referral Centres (SARCs) for better Health (MESARCH): Protocol for a one-year cohort study examining health, wellbeing and cost outcomes in adult survivors of sexual assault attending SARCs in England
AUTHORS	O'Doherty, Lorna; Carter, Grace; Lutman-White, Eleanor; Caswell, Rachel; Jackson, Louise; Feder, Gene; Heron, Jon; Morris, Richard; Brown, Katherine

VERSION 1 – REVIEW

REVIEWER	Abrahams, Naeemah South African medical Research Council, Gender and Health Unit
REVIEW RETURNED	18-Oct-2021

GENERAL COMMENTS	General Comments The article presents a protocol of a very important study. As authors indicate not much is known about the long-term health consequences following sexual assault. Doing research with sexual assault survivors are difficult and as can be seen with the pilot data from this study many survivors 'disappear' from the care systems and retention into a long term study is difficult. The addition of a qualitative component is also a good idea and the costing study is also novel and have not really been done in such detail. The attention to study ethical issues are commendable – both for survivors and staff as is the inclusion of male survivors. One of the struggles with sexual assault survivor studies is the vulnerability of the survivor soon after the event. It is therefore understanding why research team do not have a time limit since the sexual event to be enrolled. This has both pros and cons. For this study where the main outcome is PTS at 12 months after enrollment means most survivors will have varied time since event and the enrollment and assessments of PTS will also be influenced. Specific comments The authors must explain why children or at least adolescents (16-17-year-old) are not included – I am sure there is a valid reason but it needs to be explained It is a far-fetched idea that this study is a national representative study because of the huge loss to follow-up. The sample was drawn from those that sought care from sampled SARC but these SARC might not be the only services that sexual assault survivors attend in the UK. This statement must be corrected.
---

	The authors document the health impact of sexual assault and correctly indicate that not much are known about long-term health consequences. A South African study recently published their results on the long term impact of rape on HIV acquisition (Abrahams, N., S. Mhlongo, K. Dunkle, E. Chirwa, C. Lombard, S. Seedat, A. P. Kengne, B. Myers, N. Peer, C. Garcia-Moreno and R. Jewkes (2021). "Increase in HIV incidence in women exposed to rape." AIDS 35(4): 633-642.) adding to the literature. The authors provide information on what is offered at the SARC and explain the focus is on immediate care (page 7) and referral which means long term care is provided outside this system. It would be important to provide some information on what typical mental health support are provided. This is important because this study main outcome is mental health (PTS) which the SARC do not offer and might therefore not have an impact on – rather it is the mental health care and support at the referral systems or the informal systems (family friends NGOs). It is an issue for the authors to consider in their analysis. The authors must provide information on what is the standard care at the referrals sites including if it is evidence based care or best practice? Page 8 – line 11 – What is shortly after the index SARC visit? What did the pilot show . Is there a cut-off period ? Page 8 - A review of the MESARCH site shows that this study is almost a duplicate of what is presented on the site. Although there is reference to a 3 year follow-up on the site. A better more detail explanation of how this study fits into the broader program must be provided. Page 8 – line 33 – check sentence Page 8 line 48 – refence is made to de-identified data sent to the research team. Does this mean this eligibility data is not used in the study as it is not linked to a participant? This was difficult to follow. I also tried to follow this in the table but struggled. How are patient contact information collected if initial eligibility data are de-identified. At some stage patient contact data must be made available to the research staff to start enrollment and consent process. (I apologise if I missed the detail) It is noted that the authors based their sample on how many are expected to complete a 12-month follow-up interview. Nothing is said about participants who may miss a 6 month visit but are again contactable at 12 months. Will they be excluded? In reality the sample is 270/1000. The sample size to do sub analysis is a real concern and authors must be cautious as study is possible under powered (90% power) . Also, the inclusion of many covariates in the modelling will further create problems for estimation. Page 9 line 51 – what are indicators of vulnerability? ? Vulnerable to what ? Page 11 - I am surprised not to see social support scale and resilience scale – I am however aware that participants cannot be burdened with long interviews. The does the table on page 11 belong to table 2 on page 12?
--	---

	Page 13- Economic evaluation The concern her is the same as mentioned before. This economic evaluation is largely based on what happened outside of the SARCs and it is there a struggled to see how SARC model can be associated with PTS outcome if PTS management is not done within this model of care ? Essentially it will be an assessment if the SRC model refer better / have better service providers who provide better responsive care etc. (Also information is not provided on what the variations on SARC models are). It was useful to see the pilot study. It would have been useful to also see the baseline, 6 month and 12 month PTS scores for the 10 participants. Qualitative study This is a great addition s- and especially the identification of models of care. A useful tool to document the recovery journey is the life course timeline.
--	---

REVIEWER	Oliván, Barbara University of Zaragoza
REVIEW RETURNED	19-Dec-2021

GENERAL COMMENTS	Congratulations for this manuscripts which deals with a relevant topic. It is in general well-written and don't have objective errors. I have several questions or points that I expose below:  - The abstract from my point of view is not clear. The objective is not complete, neither the design (it is a lack of explanations about the qualitative study)nor the variables and instruments. - In the manuscript (las paragraph of the introduction section), the objective is not clear or complete. The objective is discovered as the article is read. From my point of view, It is is recommended that the objective is complete from the beginning of the article (from the abstract or from the introduction section) -A relevant concern I have is related the sample size. It is calculated based on the pilot work, recruitment and retention, but is it enough to analyze according to groups (by gender, vulnerablity, or different therapies, etc)? A small sample size can make logistic regressions lose power. - It is relevant to use also a qualitative methodology, and it is desirable to be shown from the beginning of the manuscript (abstract, methology design presented as mixed, explaining first quantitive methods and after the qualitative methods, etc).
---

VERSION 1 – AUTHOR RESPONSE

Reviewer: 1 Dr. Naeemah Abrahams, South African Medical Research Council Comments to the Author: General Comments: 3. The article presents a protocol of a very important study. As authors indicate not	We thank Dr Abrahams for acknowledging the value of this study.
--	--

much is known about the long-term health consequences following sexual assault. Doing research with sexual assault survivors are difficult and as can be seen with the pilot data from this study many survivors 'disappear' from the care systems and retention into a long term study is difficult. The addition of a qualitative component is also a good idea and the costing study is also novel and have not really been done in such detail. The attention to study ethical issues are commendable – both for survivors and staff as is the inclusion of male survivors.	
4. One of the struggles with sexual assault survivor studies is the vulnerability of the survivor soon after the event. It is therefore understanding why research team do not have a time limit since the sexual event to be enrolled. This has both pros and cons. For this study where the main outcome is PTS at 12 months after enrolment means most survivors will have varied time since event and the enrolment and assessments of PTS will also be influenced.	This study aims to gather health and wellbeing data from a cohort of survivors of sexual violence to inform the evidence base on impacts up to one year; however, the primary aim (driven by the original commissioned research) is to evaluate Sexual Assault Referral Centres (SARCs) in England. SARCs are an option to any survivor of sexual violence irrespective of the period of time since the trauma. Therefore, our sample needs to reflect the service users in SARC settings (from our preliminary analysis of 2500 attendees this is mainly recent assault ~60% but could be up to a year since the assault or longer, for exposure to childhood sexual abuse). The study is designed to recruit survivors soon after an 'index' visit to a SARC (we leave a period of one month for safety reasons). However, it may take longer than a month as people will vary in their vulnerability following sexual assault and their readiness to participate in research and we have put our trust in SARC and follow-on service staff to make approaches to survivors when it feels right to do so during the course of their care. We do not believe we can compromise on our approach to recruitment (i.e. avoiding cut off periods; excluding some types of sexual violence like CSA; and entrusting the task of introducing the research to support workers known to service users). However, we agree with the difficulties identified by Dr Abrahams. Thus, our intention is to account for 'time since trauma' in our analyses, these are data we have for the total pool of service users and we also have data on the nature of the assault and relationship to perpetrator. We will also know precisely the time between help seeking at SARC and enrolling in the study. All such factors can be integrated into our

	prediction models for establishing what characteristics of services could be beneficial to those seeking help.
Specific comments: 5. The authors must explain why children or at least adolescents (16-17-year-old) are not included – I am sure there is a valid reason but it needs to be explained	Thank you – yes there is a valid reason. England has a separate paediatric service offer for those aged under 18 years of age and we are conducting a separate study with young people aged 13-17 years of age. This has now been noted in the method section on page 8.
6. It is a far-fetched idea that this study is a national representative study because of the huge loss to follow-up. The sample was drawn from those that sought care from sampled SARC but these SARC might not be the only services that sexual assault survivors attend in the UK. This statement must be corrected.	We believe it would be accurate to say the study is nationally representative in terms of the inclusion of available SARC sites which we selected systematically from across England. However, we appreciate the concern and are happy to avoid such terms. The only place where we made this claim was in the article summary statements. Bullet point two has been amended accordingly. See page 3. We fully appreciate that the study is not representative of survivors of sexual violence in England. Rather our hope is to represent SARC service users. We will examine data to determine if those who enrolled in the cohort study differ from the wider population of SARC service users. Around 18,000 people attend SARC each year and we have characteristics data on 2500. We are confident this N provides a reliable picture of service users and our data will allow us to compare this with the subset that later entered the cohort study (n=337). Rather than saying our participants are nationally representative of SARC users, we will report on any differences between our enrolled participants and the 2500.
7. The authors document the health impact of sexual assault and correctly indicate that not much are known about long-term health consequences. A South African study recently published their results on the long-term impact of rape on HIV acquisition (Abrahams, N., S. Mhlongo, K. Dunkle, E. Chirwa, C. Lombard, S. Seedat, A. P. Kengne, B. Myers, N. Peer, C. Garcia-Moreno and R. Jewkes (2021). "Increase in HIV incidence in women exposed to rape." AIDS 35(4): 633-642.) adding to the literature.	Many thanks for pointing out this recent addition to the literature. We have accessed and read the article and include reference to it within our introduction on page 5.
8. The authors provide information on what is offered at the SARC and explain the focus is on immediate care (page 7)	SARCs do refer on to other organisations for ongoing support including the ISVA for advocacy and also for services that support mental

and referral which means long term care is provided outside this system. It would be important to provide some information on what typical mental health support are provided. This is important because this study main outcome is mental health (PTS) which the SARC do not offer and might therefore not have an impact on – rather it is the mental health care and support at the referral systems or the informal systems (family friends NGOs). It is an issue for the authors to consider in their analysis. The authors must provide information on what is the standard care at the referrals sites including if it is evidence based care or best practice?	health. Some SARCS provide services ‘in-house’ e.g., emotional support sessions to bridge the time waiting to access other services or ISVAs may operate out of SARCs directly – hence we are looking at the different service offers at different sites in our analyses. Where onward referral happens, the only ‘standard care’ in this context is access to the ISVA and our research thoroughly investigates participants’ experiences of ISVA services. Many of the providers are third or voluntary sector organisations, individually commissioned by local police and crime commissioners to provide ISVA care. Services users may end up going back to their GP for referrals into mental health services or they may have access to a counselling service through their university for example or a local/regional charity for supporting victims of sexual offences or domestic abuse depending on aspects of the case. Thus, we are collecting data about exactly what support our cohort members are offered and take up at SARC and beyond and agree that this needs to be considered in our analysis. We have made a concerted effort, working with statisticians for this revised version, to consider our approach to investigating the relationship between SARC factors, subsequent care, and trajectories of PTS. From there, we integrate participant level factors that may relate to PTS trajectories and then we do exploratory analysis on the potential for participant level factors to moderate the effect of care on PTS at 12 months. In terms of best practice or evidence-based care, we suggest that participants will have exposure to interventions and services with some being evidence-based (trauma focused treatments like EMDR through private, NHS and charity based psychologists and counsellors) or best practice such as the ISVA service. Much of our work is about examining the best practice and putting best evidence behind it.
9. Page 8 – line 11 – What is shortly after the index SARC visit? What did the pilot show. Is there a cut-off period?	This phrase ‘shortly after an index visit’ appears on page 6 (author page numbering) of revised manuscript in the Design and setting section of the Methods. This has been changed to “(no sooner than 4 weeks)”. The minimum timeframe was agreed in consultation with the lived experience group and staff at SARCs as it was agreed that independent sexual violence advisors would typically begin to engage with SARC service users by this time and it would likely be

	the earliest time post sexual assault occurring where survivors would be able to consider being involved in research. No cut off was ever indicated as we always anticipated recruitment would be difficult without adding further conditions. In our publication of findings, we will report the time since trauma and the time lapse between SARC visit and enrolment and expect some variation in this. We appreciate peak in PTS around 4-week post trauma so we will account for time to baseline interview in our analyses. With regards to what could be gleaned from the pilot, this was conducted at a site with integrated SARC staff and ISVA service and so the site did not demonstrate the challenges we would later face around the delays between SARC attendance and being invited to consider the research through ISVAs located in other organisations. Not having a cut off was necessary for feasibility even if not ideal from a methodological point of view. We allowed anyone aged 18 years or over who had visited a SARC since 1st August 2019 to enrol up until enrolment closed on 3rd November 2021. This detail is now included in the Sampling, recruitment and procedures section on page 8.
10. Page 8 - A review of the MESARCH site shows that this study is almost a duplicate of what is presented on the site. Although there is reference to a 3 year follow-up on the site. A better more detail explanation of how this study fits into the broader program must be provided.	On this page of the study website http://mesarch.coventry.ac.uk/whats-our-project-about/ the four distinct workstreams are outlined in lay language. WS1 - We have conducted 2 Cochrane Reviews – one quantitative and one qualitative of the evidence for effective interventions for survivors of sexual violence. These are currently under review. Protocols are previously published. WS2 - We have conducted a comprehensive mapping of SARCs across England to look at how they are set up and run, how they differ from one another and so on and this data will inform the analysis of the cohort data. The map of England SARCs is also available online. WS3 – the adult cohort study with nested qualitative study described here PLUS a young-people focused study – taking a more qualitative approach to understanding their service experiences and the

	impact of sexual violence and support on their health and wellbeing. Note that we had originally hoped to conduct a two-year follow-up of the adults in addition to the 6- and 12-month follow-up but challenges with recruitment have meant we have had to limit the study to one year and our website and other materials are currently being updated and the funder has agreed this change in Protocol V3.3. We do not believe we have referenced a three-year follow-up anywhere but will do a thorough review of our site as soon as possible. WS4 – synthesis of findings from across WS1-3. Additional detail on the wider programme of research is provided in the Design and setting section on page 6.
Page 8 – line 33 – check sentence	Because of the different page numbering generated by the peer review system versus our own it is a little difficult to ascertain which sentence requires checking. However, we have thoroughly proofread our revised manuscript and hope that clarity of expression is now optimised throughout.
Page 8 line 48 – reference is made to de-identified data sent to the research team. Does this mean this eligibility data is not used in the study as it is not linked to a participant? This was difficult to follow. I also tried to follow this in the table but struggled. How are patient contact information collected if initial eligibility data are de-identified. At some stage patient contact data must be made available to the research staff to start enrolment and consent process. (I apologise if I missed the detail)	We regret this was a difficult process to follow. We hope the clarity is improved now. To summarise, our intention was to gather data on a widespread sample of people (2500) attending SARCs, have research identification numbers assigned and which would travel with the person up to the point at which they would be referred on to our research team. At this point, we could link personal data (name and phone number, for example) with the person's ID and SARC data. It meant we had a vast de identified dataset but also an opportunity for re-linkage once we had consent. Thus, eligibility data can be used because we can link those data to all 337 participants enrolled in our study. When service user contact details come to the research team via ISVAs (usually), so too does the research identification number. The service user is made aware of this potential for re-linkage at the point they consent to their details being passed on.
It is noted that the authors based their sample on how many are expected to complete a 12-month follow-up interview. Nothing is said about participants who	No, participants will not be excluded if they miss 6-month follow-up but complete 12-month. A comment clarifying this has been added at the end of the section on 'Number of participants required'.

may miss a 6 month visit but are again contactable at 12 months. Will they be excluded? In reality the sample is 270/1000.	Our required sample size at 12 months is indeed 270 in order to have the power to conduct our analyses which are largely based around the change score for PTS between baseline and 12 months. We appreciate the limitations of our sample size but on the plus, 80% of service users we had the opportunity to speak to, joined the research study and only 4 people have actively withdrawn from the study out of 337 recruited. Our difficulty has always been a reliance on other service providers to inform service users about the research rather than approaching people ourselves. It is very useful that we have data on the population so we can compare to what extent our enrolled sample captures the characteristics of the broader population.
The sample size to do sub analysis is a real concern and authors must be cautious as study is possible under powered (90% power) . Also, the inclusion of many covariates in the modelling will further create problems for estimation.	Whilst we have maintained the assumptions for calculation of sample size, we have refined our analysis section further with the input of a statistician and emphasised the limited and exploratory nature of subgroup analysis where subgroups are very imbalanced (e.g. gender). Please see 'Number of participants required' page 8.
Page 9 line 51 – what are indicators of vulnerability? ? Vulnerable to what ?	Apologies, this is a term used by the SARCs themselves and includes factors like having a history of mental health problems or a disability. Examples have been provided in the text on page 10 to help clarify what is meant. SARCs note them as they can be relevant to providing tailored and appropriate care.
Page 11 - I am surprised not to see social support scale and resilience scale – I am however aware that participants cannot be burdened with long interviews.	We have been guided by the advice of the Lived Experience Group, study steering committee, data monitoring and ethics committee and have had to balance this with the funder's specifications and the team's vision. The authors agree that a resilience scale would have been useful to include but interviews can already run to 2 hours (sometimes we collect data over 2 sessions). However, we do have an assessment of social relationships as one of the 4 domains within the WHO QoL-Bref.
Does the table on page 11 belong to table 2 on page 12?	Something seemed to have gone awry with formatting which is hopefully now addressed in relation to table 2 on page 12.

Page 13- Economic evaluation The concern here is the same as mentioned before. This economic evaluation is largely based on what happened outside of the SARCs and it is there a struggle to see how SARC model can be associated with PTS outcome if PTS management is not done within this model of care? Essentially it will be an assessment if the SRC model refer better / have better service providers who provide better responsive care etc. (Also information is not provided on what the variations on SARC models are).	The reviewer has raised some of the central challenges for this work which is about evaluating an early step in a pathway of care for sexual violence survivors. As we recognised the importance of the full pathway, we have ensured to collect data on service use and perceived benefit/harm related to SARC, and in relation to all follow on sexual violence services and mental health care accessed by service users. We agree that the primary analysis will be about the effectiveness of the SARC service delivery and this is what we have set out. In response to your important points, we have revised our analysis section to make more explicit our appreciation of the role of not only the SARC but also the follow-on care (anything related to the index sexual assault) which is assessed over the course of the study. We have added further detail on what variations in SARC models we would plan to examine. Please see reference to SARC-level characteristics in 'Data analysis' > Last paragraph page 13
It was useful to see the pilot study. It would have been useful to also see the baseline, 6 month and 12-month PTS scores for the 10 participants.	Please note that the pilot study only involved one 3-month follow-up. There are no 6- or 12-month scores to report. If the reviewers' feel strongly that these would add value we could look them up, but the main purpose of the pilot study was to test and refine the recruitment strategy. We are not convinced the baseline and 3-month follow up PTS scores for ten participants is particularly informative.
Qualitative study This is a great addition s- and especially the identification of models of care. A useful tool to document the recovery journey is the life course timeline.	Thanks for this helpful suggestion – we will look into this.
Reviewer: 2 Dr. Barbara Oliván, University of Zaragoza Comments to the Author: Congratulations for this manuscripts which deals with a relevant topic. It is in general well-written and don't have objective errors.	Many thanks for your positive assessment of our protocol paper.
I have several questions or points that I expose below: - The abstract from my point of view is not clear. The objective is not complete,	We agree that more detail would be useful, but we are limited to 300 words. We have made small refinements in the text which we hope improve the clarity.

neither the design (it is a lack of explanations about the qualitative study) nor the variables and instruments.	
- In the manuscript (last paragraph of the introduction section), the objective is not clear or complete. The objective is discovered as the article is read. From my point of view, It is recommended that the objective is complete from the beginning of the article (from the abstract or from the introduction section)	We have also refined the last paragraph and hope the objectives are now clear: This mixed-methods cohort study is the first to consider the health outcomes and cost trajectories of those who attend SARC in England. It aims to evaluate the relationship between SARC factors, subsequent care (follow on specialist sexual assault services and mental health care), and trajectories of PTS; the relationship between participant characteristics and trajectories of PTS; and finally, the role that participant characteristics may have in moderating the benefits of different SARC factors and subsequent care. It represents the first such study to consider the broader sexual health and wellbeing of survivors attending SARCs alongside other assessments of their health and mental wellbeing over time. We also report here how findings from our pilot study at one English SARC and our early data collection experiences have informed a revised research protocol.
A relevant concern I have is related the sample size. It is calculated based on the pilot work, recruitment and retention, but is it enough to analyze according to groups (by gender, vulnerability, or different therapies, etc)? A small sample size can make logistic regressions lose power.	Thanks for this – we have consulted with our statistician and made amends to the relevant text within the manuscript – please see pages 13-14.
- It is relevant to use also a qualitative methodology, and it is desirable to be shown from the beginning of the manuscript (abstract, methodology design presented as mixed, explaining first quantitative methods and after the qualitative methods, etc).	We have added the term mixed-methods at the beginning of the methods and analysis section of the abstract and the term quantitative in relation to first describing these assessments to help clarify that this is a mixed methods study. The same has been done in the main body of the paper.

VERSION 2 – REVIEW

REVIEWER	Oliván, Barbara University of Zaragoza
REVIEW RETURNED	02-Feb-2022
GENERAL COMMENTS	The manuscript from my point of view is interesting, clear, without objective errors and the authors have improved the quality of it.

	The authors have change the authorship of the manuscript and I suppose the authors will have done as the journal indicates.
--	---